# Serious Life Events in People with Visual Impairment Versus the General Population

**DOI:** 10.3390/ijerph182111536

**Published:** 2021-11-02

**Authors:** Audun Brunes, Trond Heir

**Affiliations:** 1Section for Trauma, Catastrophes and Forced Migration—Adults and Elderly, Norwegian Centre for Violence and Traumatic Stress Studies, 0484 Oslo, Norway; trond.heir@medisin.uio.no; 2Institute of Clinical Medicine, Faculty of Medicine, University of Oslo, 0318 Oslo, Norway

**Keywords:** accidents, blindness, assaults, prevention, serious life events, visual impairment

## Abstract

The present study aimed to examine the lifetime exposure to serious life events in people with visual impairment compared with the general population. Data were derived from a telephone survey including a probability sample of 736 adults with visual impairment (response rate: 61%). The lifetime prevalence of direct experiences with seventeen different categories of serious life events (Life Events Checklist for DSM-5 (LEC-5)) were compared to that obtained from the general Norwegian population (*N* = 1792, 36% response rate). Altogether, 68% of people with visual impairment had been directly exposed to at least one serious life event, with equal rates among males and females (*p* = 0.59). The prevalence of serious life events was higher than for the general population (60%, *p* < 0.001), especially for fire or explosions, serious accidents, sexual assaults, life-threatening illness or injury, and severe human suffering. In conclusion, our results indicate that people with visual impairment are more prone to experiencing serious life events. This highlights the need for preventive strategies that reduce the risk of serious life events in this population.

## 1. Introduction

Serious life events, such as traffic accidents, physical assaults, and rape, are events of such a serious nature that they can evoke major psychological distress, including feelings of intense fear and helplessness, in almost anyone [1]. For people with visual impairment, the absence of vision may affect the individual’s experience and response to a life event [2], which may amplify the stressfulness of the situation [3]. In fact, serious life events have been highlighted as an important barrier for the fulfillment of human rights in this population [4], and coincided with a variety of adverse health outcomes, including posttraumatic stress disorder (PTSD) [2,3], substance abuse [2], depression [5,6], loneliness [7], and all-cause mortality [8]. To design preventive health strategies and facilitate the basic human needs of people with visual impairment, precise knowledge about the extent of serious life events is essential.

Visual impairment is a heterogeneous condition affecting about 1 billion, or 13 percent, of the world’s population [9]. The prevalence is highest in low-income countries, but also substantial in Western Europe and many other high-income countries [10]. For people with visual impairment, a lack of visual information and the group’s living situation could make them more susceptible to experiencing serious life events [11,12]. For example, studies which have addressed exposure to specific types of events have shown that people with visual impairment have a similar or higher prevalence of accidents [2,13], violence [14,15,16], abuse [17], and sexual assaults [16,18], compared to others. On the other hand, there are contrasting hypotheses suggesting that people with visual impairment may have a lower risk of some types of serious life events [19]. This could in part be due to the isolated and socially withdrawn life that many of them lead [7].

Knowledge about the extent of serious life events in the visual impairment population is limited. In two systematic reviews of the literature [2,3], we identified one study that examined a broad range of serious life events in people with visual impairment (search methods described in the Appendix A). In that study, which was from a war zone in Lebanon, people with vision or hearing loss had experienced less serious life events than their sighted or hearing peers [19]. To contribute to the literature, the present study aimed to examine the lifetime prevalence of direct exposure to serious life events among people with visual impairment compared to the general population.

## 2. Materials and Methods

### 2.1. Visual Impairment Population

An anonymous survey was administered to a large probability sample of adult members of the Norwegian Association of the Blind and Partially Sighted. All members aged 18 years or older were eligible to participate if they had a diagnosis of visual impairment and were able to speak and understand the Norwegian language. Data were collected between January 2017 and May 2017 through structured telephone interviews. Most members were of older age. To involve the entire visual impairment population, we therefore used age-stratified sampling. First, the study population was divided into four age groups (years: 18–35, 36–50, 51–65, ≥66) and then we randomly surveyed an equal number of members across the different age groups. Of the 1216 members we contacted, 736 (61%) participated by completing the interview. A flow chart of the sample selection is provided elsewhere [20].

### 2.2. General Population

We extracted norm data on baseline characteristics and serious life events in the general population from the Norwegian Population Study (NorPop) [21]. The study included a nationwide probability sample of adults aged 18 years or older. Data were collected between 2014 and 2015 by self-administered postal questionnaires. Of the 5500 eligible participants, 9 persons had died, 21 were not able to fill out the questionnaire, and 499 inquiries had non-valid addresses. Of the remaining 4971 inquiries, 1792 individuals (36%) responded by completing and returning the questionnaire. NorPop was approved by the Regional Committee for Medical and Health Research Ethics.

### 2.3. Measures

#### 2.3.1. Background Information

Both surveys collected data about the participant’s age, sex, place of residence (urban, rural), education (years: <11, 11–13, ≥14), work status (unemployed, employed, retired), and marital status (married/cohabitant, unmarried). In the visual impairment population, we included self-reported information about the degree of visual impairment (blindness, moderate-to-severe impairment), nature of vision loss (congenital, childhood (1–24 years), or adulthood (≥25 years)), cause of vision loss (congenital, disease, or injury), and having additional impairments (no, yes).

#### 2.3.2. Serious Life Events

In both surveys, the Life Events Checklist for DSM-5 (LEC-5) was included to assess the participant’s personal experiences with serious life events. LEC-5 screens for sixteen different categories of serious life events (e.g., fire or explosions, traffic accidents, serious accidents at home, work or during leisure time, physical or sexual assaults, war combat, and life-threatening illness or injury), as well as other stressful life events not specified in the list. In the visual impairment population, we added one more event category: illness or injury causing vision loss, since this has proven to be a unique traumatic event in this population [22]. Previous versions of the LEC-5 have been shown to be reliable and valid in a variety of study populations [23].

### 2.4. Statistical Analyses

We used Stata Version 16 (StataCorp, College Station, TX, USA) for all statistical analyses. The significance level was set at *p* = 0.05. We calculated the lifetime prevalence of overall and category-specific exposure to serious life events for the visual impairment population and the general population. The estimates were presented for the total sample in each population and, if applicable, separately for males and females. Comparisons of categorical variables were performed using Pearson’s chi-squared tests.

In supplementary analyses, we performed unadjusted and fully adjusted Poisson generalized linear models with a log-link function and a robust variance estimator to examine independent variables (sociodemographic and vision-related factors) and their associations with each category of serious life events [24]. The results were presented as prevalence ratios (PRs) and 95% confidence intervals (CIs). The supplementary analyses were only conducted for those life events being significantly different between the visual impairment population and the general population and having a prevalence of greater than 10% in the visual impairment population. To reduce the risk of sparse data bias, we chose to dichotomize the education and work status variables in the regression models.

## 3. Results

### 3.1. Sample Characteristics

The main analyses included 736 adults with visual impairment and 1792 adults from the general population. In both surveys, non-participants were more likely than participants to be of young or old age. The visual impairment population had no sources of missing data among the participants, whereas the percentage of missing data in the general population ranged between 0% and 2% across the different variables.

Table 1 shows characteristics of males and females from the visual impairment population and the general population. Male and female participants with visual impairment had lower levels of education, were more unemployed, and to a lesser extent married compared to the general population. In the visual impairment population, the onset-age of vision loss ranged from 0 to 76 years (mean: 19 years), and was primarily caused by diseases (50%), followed by congenital causes (43%), and injuries (7%). A total of 25% had self-reported blindness, and the remaining 75% had self-reported moderate-to-severe impairment. Roughly one in three reported other impairments in addition to their vision loss.

### 3.2. Lifetime Prevalence of Serious Life Events

After excluding ‘illness or injury causing vision loss’ from the analysis to obtain equal event categories in the two populations, more people from the visual impairment population reported that they had been directly exposed to at least one serious life event during their lifetime (67.5%, 95% CI: 64.0–70.9) compared to people in the general population (59.8%, 95% CI: 57.6–62.1) (*p* < 0.001).

As displayed in Figure 1, people with visual impairment had a greater exposure to a broad range of event categories compared to the general population. Particular large differences were observed for personal experiences of fire or explosion, serious accidents happening at work, home, or during leisure time (e.g., fall accident), exposure to toxic substances, sexual assaults, war events, life-threatening illness or injury, and severe human suffering (each *p* < 0.05). There was no single category of life events of which the general population had higher rates of exposure than the visual impairment population.

In the visual impairment population, the three most common life events were illness or injury causing vision loss (36%), life-threatening illness or injury (25%), and traffic accidents (23%). Males and females had equal rates of exposure to serious life events overall (*p* = 0.59). However, sex differences were observed for specific event categories. Males were more likely than females to report direct exposure to toxic substance and life-threatening illness or injury. In contrast, females were more often exposed to sexual assaults, other unwanted sexual experiences, and other stressful events (Appendix A).

### 3.3. Supplementary Analyses

#### 3.3.1. Fire or Explosion

Past exposure to fire or explosions were higher for those who had acquired their vision loss at some point in life compared with those with congenital vision loss (Appendix A).

#### 3.3.2. Serious Accidents at Work, Home, or during Leisure Time

Young age and having other impairments were associated with a greater lifetime exposure to serious accidents happening at work, home, or during leisure time (Appendix A).

#### 3.3.3. Sexual Assaults

Females represented most of those who had experienced a sexual assault. Moreover, direct exposure to sexual assaults were higher among those who were of young age, were unemployed and unmarried, and had other impairments in addition to the vision loss (Appendix A).

#### 3.3.4. Life-Threatening Illness or Injury

In an adjusted regression model, life-threatening illness or injury was strongly associated with having lost vision late in life and having other impairments in addition to the vision loss. The prevalence was also higher among male participants and those who were blind and of older age (Appendix A).

#### 3.3.5. Severe Human Suffering

Participants who lost their vision late in life were more likely to report personal experiences of severe human suffering than those with congenital vision loss. Severe human suffering was also more prevalent in those with other functional impairments (Appendix A).

## 4. Discussion

In this cross-sectional study, we found that people with visual impairment were more prone to serious life events compared to the general population. This was especially true for direct experiences of fire or explosion, serious accidents, sexual assaults, life-threatening illness or injury, and severe human suffering.

### 4.1. Strengths and Limitations

Strengths of this study include the large probability sample of adults with visual impairment, the use of a validated instrument in the assessment of serious life events, the possibility to obtain robust estimates across different subgroups of the population, and the inclusion of a probability sample from the general population.

Our study had certain limitations. First, the representativeness of our sample is questionable as it was recruited from a member organization for the blind and partially sighted. Compared to 2015 census data from Statistics Norway including people with self-rated visual difficulties [25], our study sample did not differ in terms of sex, employment, and place of residence, but had a higher level of education. Furthermore, the rate of blindness was higher in our study compared to that reported previously [26]. Second, we cannot rule out that there were people who were blind or visually impaired in the general population data. In that case, the real difference between the visually impaired and sighted is expected to be greater than our estimate. Third, we limited gender to two sexes, male and female, which may be criticized for not including transgendered or non-binary people. Fourth, the use of self-reports on serious life events may have affected the validity of the estimates. For example, the retrospective reports of serious life events may lead to recall bias. Some events could be forgotten or no longer considered important, whereas others could have been subjected to memory amplification [27]. Fifth and last, non-participation may have introduced biased prevalence estimates in both study populations. We have limited information about the non-participants and do not know how non-responding might have influenced our results.

### 4.2. Interpretation and Comparison with Other Studies

Our findings of a high prevalence of serious accidents among people with visual impairment are consistent with previous studies [2,13]. The risk may be linked to the importance of vision to detect, avoid, or flee from potential high-risk situations [11,12]. For example, in a qualitative study of blind and partially sighted people, we found that most of the accidents reported (e.g., falling or fire accidents) were directly attributed to the vision loss itself [11]. The risk of accidents may be further exacerbated by a lack of universal design of environments and restricted use of needed assistive aids [11,28]. A reversed causality, i.e., accidents that are the cause of vision loss, does not seem to explain our findings, since the prevalence of accidents remained similar after excluding those who reported injuries as the main cause of vision loss from the analyses (results not shown).

Although previously published [29], we have included results on sexual assaults to provide a comprehensive picture of the broad array of serious life events that people with visual impairment experience. The high occurrence of sexual assaults may be related to a vulnerability in being a part of a marginalized group. Assaults are largely about power and oppression [30], and low socio-economic status or being more prone to social isolation or dependency are important risk factors [31].

The findings of a high prevalence of life-threatening illness or injury agrees with the explanation of life-threatening illness or injury as the main cause of vision loss, or more plausible, that the risk of serious illness or injury is linked to low socio-economic status or an unhealthy lifestyle. Hence, factors being more prevalent in people with visual impairment relative to the general population [8,26]. Uniquely for people who are blind or partially sighted is the fact that they have been exposed to injury or illness that resulted in a vision loss. Although this has not been a life-threatening event, vision loss can be considered a threat to a person’s sense of independence, safety, and control [32], and subsequent stress reactions, such as flashbacks, hyperarousal, and avoidance of reminders, are common [23].

The high prevalence of severe human suffering is a warning that many people with visual impairment may have experienced great difficulties in life. This was more common for people who have lost their vision late in life, possibly due to less adaptability, and people who had other impairments in addition to their vision loss, probably due to poor health or a greater functional or social implication of the health condition itself. High burden of mental disorders, such as depression or substance use [2,5,6], should be included as possible explanations for the high rates of suffering.

### 4.3. Implications

Higher exposure to serious accidents happening at work, at home, or during leisure time, indicates a need for better adaption of the physical environment for people with visual impairments. This means a universal design of physical spaces with an emphasis on safety and ease of use for visually impaired people. Appropriate adaptations must apply to housing, schools, workplaces, leisure activities, the transport sector, and public areas. Accident prevention in this population may also encompass efforts which facilitate the use of assistive aids in situations where it can increase the individual’s safety.

Sexual assault prevention includes interventions that raise public awareness and upgrade professional education about the vulnerability of specific groups such as children and adults with visual impairments [33]. Ensuring social equalization and integration of people with impairments should be a key objective in making them more resilient to discrimination, stigmatization, and sexual assaults. Universal design of information and physical spaces will create more equal conditions regardless of visual impairments and help to promote self-confidence and independence in the individual.

The extent of exposure to life-threatening illness or injury as well as severe human suffering is largely about health and social positions in society, including access to health care services and social benefits. From a minority perspective, it is important to acquire more knowledge about the specific factors related to being blind or partially sighted, and how social and structural conditions can be changed for the better for these people. Ultimately, however, the promotion of social equalization for the benefit of people who are visually impaired is a matter of political intervention.

## 5. Conclusions

In conclusion, our results indicate that people with visual impairment are more prone to serious life events than the general population. A higher prevalence of accidents requires adaptation of indoor and outdoor environments through principles of user-friendliness, safety, and universal design. Social equalization and integration of people with visual impairments may be a main objective in making them less vulnerable to serious life events in general.

## Figures and Tables

**Figure 1 ijerph-18-11536-f001:**
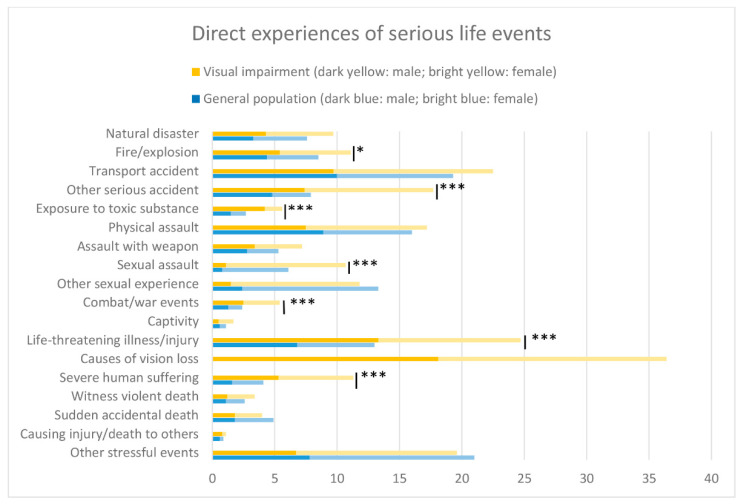
Direct exposure to serious life events in people with visual impairment (*N* = 736) compared with the general population (*N* = 1792). * *p* < 0.05, *** *p* < 0.001.

**Table 1 ijerph-18-11536-t001:** Study characteristics of males and females from the visual impairment population (*N* = 736) and the general population (*N* = 1792).

	VI Male (*N* = 333)	GP Male (*N* = 834)		VI Female (*N* = 403)	GP Female (*N* = 945)	
Age, mean (s.d.) Education, *n* (%) <11 years	51.1 (17.0)	55.7 (15.9)	*p* < 0.001	51.7 (17.3)	51.0 (17.0)	*p* = 0.48
			*p* = 0.003			*p* < 0.001
	46 (13.8)	62 (7.5)		69 (17.1)	79 (8.4)	
11–13 years ≥14 years	124 (37.2)	336 (40.5)		162 (40.2)	346 (36.7)	
	163 (49.0)	432 (52.0)		172 (42.7)	517 (54.9)	
Work status, *n* (%) Employed # Unemployed Retired			*p* < 0.001			*p* < 0.001
	152 (45.7)	526 (63.4)		143 (35.5)	641 (68.3)	
	100 (30.0)	60 (7.2)		148 (36.7)	72 (24.1)	
	81 (24.3)	244 (29.4)		112 (27.8)	28 (7.7)	
Marital status, *n* (%) Married/cohabitant			*p* < 0.001			*p* < 0.001
	166 (49.9)	634 (76.3)		181 (44.9)	647 (68.9)	
Unmarried §	167 (50.1)	197 (23.7)		222 (55.1)	292 (31.1)	
Place of residence, *n* (%)			*p* = 0.32			*p* = 0.003
Rural areas	172 (51.7)	399 (48.4)		227 (56.3)	444 (47.3)	
Urban areas	161 (48.4)	426 (51.6)		176 (43.7)	494 (52.7)	

Notes: VI: visual impairment; GP: general population; #: the employed category encompassed people reporting to be in work, under education, or in military service; §: unmarried involved those who were not married, divorced, or widowed.

## Data Availability

Data are from the research project European Network for Psychosocial Crisis Management—Assisting Disabled in Case of Disaster (EUNAD). According to the approval from the Norwegian Regional Ethical Committee, data are to be stored properly and in line with the Norwegian privacy protection laws. Data contains sensitive information from a small group. Public availability may result in the possibility of indirect identification, and thus would compromise privacy of the participants. However, the data are freely available to interested researchers upon request, pending ethical approval from our ethical committee: post@helseforskning.etikk.no. The project leader, Prof. Trond Heir (trond.heir@medisin.uio.no), may also be contacted with requests for the data underlying our findings.

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
