# Peer review of "Serious Life Events in People with Visual Impairment Versus the General Population"

_ijerph, 2021, doi:10.3390/ijerph182111536_

Round 1

Reviewer 1 Report

Dear Authors

It is very interesting research that confronts an important issue. The paper itself, however, would benefit from some clarifications, if not improvements. 

  1. Line 49: The readers would appreciate some details regarding the systematic review you mentioned since it's at least three years old: what databases were searched, using what keywords, in what language(s), etc., whether an attempt was made to update it. Since 2018 you have published some papers yourselves.
  2. Line 63 and 74: The reference data regarding the general population were collected in 2014 and 2015 while your survey was conducted in 2017. Combined with references [3] and [9], it leaves the impression that at least part of this research was published before.
  3. Line 81 and Table 1, line 193 and section 4.1: Since the article addresses significant difficulties of a particularly vulnerable group, reducing gender to two sexes, male and female, is a surprise. I'm guessing it may be related to the fact that the survey regarding the general population was conducted this way. In line 193, you mentioned your research had certain limitations. I believe it's one of them. 
  4. Lines 212-213, line 262: Speaking of limitations, there are ways to normalize the statistics, especially in the case of surveys made on groups difficult to compare. How about using one of them? How many people in Norway suffers from this sort of impairment? What is the percentage in other European countries? Are there other vulnerable groups that face similar issues?
  5. Lines 242-243: universal design has been a standard for at least a couple of decades. Unfortunately, the answer is not that simple. 
  6. Lines 265-267: I would expect less obvious a conclusion.

Last but not least, the abstract and keywords are not the strongest points of the paper and should be revised. Given the interdisciplinary profile of the journal, it offers too many details instead of painting a slightly bigger picture. I encourage you to present the results of your research in a wider context in the final section.

I hope you'll find my comments helpful. I believe that, with your experience, turning this paper in its even better version will not be a problem.

Reviewer 2 Report

The sentence The findings that people with visual impairment are more prone to serious life events highlight the need for preventive strategies is not a result of the study statistics.

I proceed to make the comments that he tells me:
1. In my opinion, the strengths of the manuscript are:
- Large sample of participating subjects.
- Good description and contextualization.
- Adequate structuring in categories of results
- Good final discussion and significance of results

And the weaknesses are:
- Extension of the conclusions

  2. My main recommendations for the
  improvement of the manuscript would be:
Expand the conclusions and relate them to the research objectives, better specified

Reviewer 3 Report

The study deals with the little existing knowledge about the risk of serious life events in people with visual impairment. It aims to describe lifetime exposure to serious life events in people with impairment compared to the general population.
The subject is of interest and the elaboration adapts to the structure of the magazine. It presents an adequate methodological approach, a good theoretical introduction and a clear presentation of results and conclusions.

Round 2

Reviewer 2 Report

Very descriptive and scientific methods used